# Vitamin D Deficiency and Risk Factors Related to Acute Psychiatric Relapses in Patients with Severe Mental Disorders: A Preliminary Study

**DOI:** 10.3390/brainsci12080973

**Published:** 2022-07-24

**Authors:** Michele Fabrazzo, Salvatore Agnese, Salvatore Cipolla, Matteo Di Vincenzo, Emiliana Mancuso, Antonio Volpicelli, Francesco Perris, Gaia Sampogna, Francesco Catapano, Andrea Fiorillo, Mario Luciano

**Affiliations:** Department of Psychiatry, University of Campania “Luigi Vanvitelli”, 80138 Naples, Italy; agnesesalvatore@gmail.com (S.A.); salvatore2211@gmail.com (S.C.); dr.matteodivincenzo@gmail.com (M.D.V.); mancuso.emiliana@gmail.com (E.M.); volpicelliantonio@hotmail.it (A.V.); francesco.perris@unicampania.it (F.P.); gaia.sampogna@gmail.com (G.S.); francesco.catapano@unicampania.it (F.C.); andrea.fiorillo@unicampania.it (A.F.); mario.luciano@unicampania.it (M.L.)

**Keywords:** vitamin D deficiency, parathyroid hormone, psychiatric relapses, affective disorders, psychotic disorders

## Abstract

Previous studies have indicated that vitamin (Vit) D deficiency is frequent in psychiatric patients, regardless of diagnostic category. We aimed to assess whether acute psychiatric relapses in inpatients was associated with Vit D deficiency compared to stabilized outpatients. The cohort (152 total patients, 75 males and 77 females) had a mean age of 47.3 ± 14.4 years at admission and was grouped according to psychiatric diagnosis. Psychopathological symptom severity was assessed by the Brief Psychiatric Rating Scale (BPRS), a multidimensional symptom inventory. Total calcium serum levels were measured using standard laboratory methods, while plasma levels of 25-OH-Vit D and parathyroid hormone (PTH) were measured by automated chemiluminescence immunoassays. The psychiatric inpatient subgroup showed a significant difference in serum levels of 25-OH-Vit D and PTH (*p* < 0.001). Correlation analysis between serum levels of 25-OH-Vit D and BPRS total and subitem scores indicated a significantly negative relationship. In addition, linear regression analysis evidenced that the inpatient condition might predict low PTH and 25-OH-Vit D serum levels. Hospitalized psychiatric patients are at increased risk for Vit D deficiency regardless of their diagnostic categories. The mechanism underlying the association between acute psychiatric relapses and Vit D deficiency remains unclear. Therefore, screening for Vit D deficiency should pertain to the health assessment of patients with major psychiatric disorders.

## 1. Introduction

Over the past twenty years, several studies have provided convincing evidence that vitamin (Vit) D plays a crucial role in calcium homeostasis and bone health [1,2]. Furthermore, Vit D has proved to be involved in numerous brain processes, including neuroimmunomodulation, neuroprotection, and neuroplasticity as well as brain development, modulated biosynthesis of neurotransmitters, and neurotrophic factors [3]. Recent studies have also shown that most tissues and cells of the human body, including the brain, have Vit D receptors, suggesting that Vit D might be implicated in numerous other functions [4], such as the modulation of inflammatory responses, which is often dysregulated in patients with severe mental disorders [5,6,7].

In the general population, Vit D deficiency has been associated with an increased risk of cardiovascular diseases (CVDs) and related disorders such as stroke, hypertension, and heart failure [8], which are often over-reported in patients with severe mental disorders compared to the general population [9,10,11,12]. In addition, low plasma Vit D levels appear to relate to a few health risk factors for CVDs, such as high body mass index (BMI) [13], larger waist circumference [14], increased rate of smoking, and drug abuse [15,16,17]. Obesity, metabolic syndrome, type 2 diabetes, and CVDs are frequent disorders that might worsen the condition of psychiatric patients, which is already burdened by medications′ side effects, genetic vulnerability, and unhealthy lifestyles [18,19,20].

The association between Vit D deficiency and psychiatric disorders has raised critical attention. Accordingly, several studies have indicated an increased risk for Vit D deficiency in psychiatric patients with psychotic and affective spectrum disorders [21,22,23]. For example, Belveri Murri et al. (2013), in a meta-analysis of studies evaluating Vit D deficiency in psychiatric patients, along with other authors, reported that patients with psychotic spectrum disorders are more likely to suffer from Vit D deficiency when compared with healthy controls and patients with other psychiatric disorders [24]. However, the validity of such findings needs to be more precisely assessed in patients diagnosed with psychiatric disorders other than schizophrenia and related disorders, such as mood disorders, autistic spectrum disorder, obsessive-compulsive disorder, and Alzheimer′s disease [4,25,26,27].

Low plasma Vit D levels are reported in patients with established and first-episode psychosis [28,29]. However, to what extent hypovitaminosis D is related to disease mechanisms or might influence health outcomes among such a population of psychiatric patients remains unexplained. Accordingly, the association between Vit D deficiency and psychotic features requires extensive investigation to determine whether Vit D deficiency is a mediator of illness severity, a result of illness severity, or both. Furthermore, psychiatric patients, especially those affected by schizophrenia, frequently do not follow a healthy lifestyle. Symptoms exacerbation might lead to severe food restrictions in some patients, contributing to worsening Vit D deficiency and psychopathological status [30].

In light of the above considerations and given the great variability of results from several clinical studies reporting a low serum concentration of Vit D in patients with different mental illnesses, we aimed to examine whether acute relapses of psychiatric disorders, regardless of diagnosis, was associated with Vit D deficiency in comparison with a control group of outpatients in stabilized psychopathological and psychopharmacological conditions.

## 2. Materials and Methods

### 2.1. Participants

In our cross-sectional cohort study, we recruited patients at the Department of Psychiatry of the University of Campania “Luigi Vanvitelli” from June 2019 to June 2021. Eligibility criteria to be enrolled were: (a) age between 18 and 65 years, (b) a psychiatric diagnosis according to the Diagnostic and Statistical Manual of Mental Disorders—fifth edition (DSM-5) [31], and (c) willingness to participate in the study. Exclusion criteria were: (a) inability to provide written consent to participate in the study, (b) moderate or severe cognitive impairment, (c) comorbidity with a neurologic disease or drug and/or alcohol abuse, (d) being pregnant or in the post-partum period, and (d) currently being treated with medications that may alter the calcium metabolism. All patients gave their written informed consent to participate in the study after receiving a complete description of the aims and design of the study. The study was carried out according to the latest version of the Declaration of Helsinki and approved by the Ethics Committee of the University of Campania “Luigi Vanvitelli” (no: N001567/28.01.2018).

### 2.2. Procedures and Measures

#### 2.2.1. Sociodemographic and Clinical Characteristics

Patients′ clinical and sociodemographic characteristics, including gender, age at study entry, employment status, educational level, family history of psychiatric illnesses, lifetime number of hospitalizations and relapses, the pattern of illness symptomatology, primary psychiatric diagnoses, and the number of suicidal attempts, were recorded in an ad hoc schedule. BMI was calculated for each participant and according to the World Health Organization (WHO) criteria (2000). Subjects were classified as normal weight (BMI 18.50–24.99 kg/m^2^), overweight (BMI 25.00–29.99 kg/m^2^), obese (BMI ≥ 30.00 kg/m^2^), and underweight (BMI < 18.50) [32].

Psychopathological symptom severity was assessed by the Brief Psychiatric Rating Scale (BPRS, 18-item version), a multidimensional symptom inventory [33,34]. The BPRS scores were used as approximate measures of psychiatric illness severity. The scale consists of 18 items that measure the following symptoms: (1) somatic concern, (2) anxiety, (3) emotional withdrawal, (4) conceptual disorganization, (5) guilt feelings, (6) tension, (7) mannerisms and posturing, (8) grandiosity, (9) depressive mood, (10) hostility, (11) suspiciousness, (12) hallucinatory behavior, (13) motor retardation, (14) uncooperativeness, (15) unusual thought content, (16) blunted affect, (17) excitement, and (18) disorientation. Items were rated on a seven-item Likert-type as follows: 0 = “not assessed”, 1 = “not present”, 2 = “very mild”, 3 = “mild”, 4 = “moderate”, 5 = “moderately severe”, 6 = “severe”, and 7 = “extremely severe”. Several studies have tested BPRS psychometric properties and reported an inter-rater reliability of BPRS items ranging from 0.56 (tension) to 0.87 (guilt feelings and hallucinatory behavior) as well as an inter-rater reliability for overall scores ranging from 0.67 to 0.95. Moreover, convergent reliability appeared up to 0.93 [35,36].

#### 2.2.2. Assessment of Serum Parameters

Serum levels of total calcium were measured using standard laboratory methods, and plasma levels of Vit D (25-OH-Vit D) and PTH were measured by automated chemiluminescence immunoassays (DiaSorin Liaison^®^, Saluggia, VC, Italy, Siemens ADVIA Centaur^®^ XP). Blood samples from all patients were collected at recruitment in the morning, adopting a standard procedure. Immediately after, blood samples were gathered and sent to the University of Campania “Luigi Vanvitelli” laboratory. The normal reference range for total serum calcium was 8.6–10.2 mg/L, 4.6–58.1 pg/mL for PTH, and >30 ng/mL for 25-OH-Vit D. According to the WHO, we defined Vit D deficiency as serum 25-OH-Vit D below 20 ng/mL (50 nmol/L) [37]. Specifically, Vit D levels were categorized according to the serum concentration as “severely deficient” (0–10 ng/mL), “deficient” (11–20 ng/mL), “insufficient” (21–32 ng/mL), and “adequate” (33–49 ng/mL) [38].

#### 2.2.3. Statistical Analyses

Descriptive statistics were calculated for sociodemographic and clinical characteristics variables and based on relevant assessment instruments. As appropriate, data were presented as means and standard deviations (sd) or frequencies and percentages (%). The Kolmogorov–Smirnov test was adopted to check the normality of our samples′ distribution. Correlation analysis was performed to explore the association of serum levels of PTH, 25-OH-Vit D, and calcium with continuous variables. The Student′s *t*-test for independent samples was performed to assess the association of serum levels of PTH, 25-OH-Vit D, and calcium with discrete variables. Linear regression analysis was performed using serum PTH, 25-OH-Vit D, and calcium as dependent variables. Independent variables were selected among those with a positive association in the univariate analysis and those considered relevant in the scientific literature. The level of statistical significance was set at *p* < 0.05. Statistical analysis was performed with the Statistical Package for Social Sciences, version 21 (SPSS, Chicago, IL, USA).

## 3. Results

### 3.1. General Characteristics of the Enrolled Patients

Sociodemographic, clinical, and biochemical characteristics of the 152 patients enrolled (75 males and 77 females) are reported in Table 1 along with those featuring inpatient and outpatient subgroups.

A total of 71 patients (47.0%) of the whole cohort had a partner, and 48 (32.0%) were employed. Furthermore, the number of hospitalizations in patients′ past medical history was significantly higher in the inpatient subgroup (*p* < 0.001), while the number of previous relapses was greater in the outpatient subgroup (*p* < 0.001).

In addition, 67 patients (48.9%) had a positive family history of psychiatric disorders, and 122 patients (80.3%) had received a diagnosis of mood disorder, which was significantly higher in the outpatient subgroup (*p* < 0.001). Specifically, the diagnosis was of bipolar disorder (n = 78, 48.1%) and major depressive disorder (n = 44, 32.2%).

Furthermore, 25 patients (16.0%) had received a diagnosis of a psychotic disorder, namely detail schizophrenia (n = 18, 11.8 %) and schizoaffective disorder (n = 7, 4.2%), and were assigned to the inpatient subgroup, which also included patients with other diagnoses, such as obsessive-compulsive disorder (n = 5, 3.3%).

BPRS total score analysis evidenced that the inpatient subgroup had a significantly higher mean total score (*p* < 0.001) along with depression, thought disorders, activity, and hostility/suspiciousness item scores when compared to the outpatient subgroup (Table 1).

In the total cohort, serum levels were 30.6 ± 20.1 pg/mL for PTH, 23.1 ± 13.4 pg/mL for 25-OH-Vit D, and 9.3 ± 6.4 mg/dL for calcium. In the outpatient subgroup, a significantly higher serum concentration was reported for PTH (38.5 ± 24.4 pg/mL) and 25-OH-Vit D (28.3 ± 15.0 pg/mL) when compared to values of the inpatient subgroup (*p* < 0.001) (Table 1). No patients of in- and outpatient cohorts were taking oral Vit D at admission.

### 3.2. Univariate Analyses

Correlations between serum biochemical parameters and clinical variables are reported in Table 2.

Serum 25-OH-Vit D levels emerged as negatively correlated with BMI (*p* < 0.01), BPRS total (*p* < 0.01), and subitem scores, namely anxiety/depression (*p* < 0.05), anergy (*p* < 0.01), thought disorders (*p* < 0.01), activity (*p* < 0.05), and hostility/suspiciousness (*p* < 0.05).

### 3.3. Multivariate Analyses

The results of the linear regression analysis are listed in Table 3. The model was run to assess the independent predictors associated with the PTH, calcium, and 25-OH-Vit D serum levels and adjusted for gender, age at onset and admission, marital status, educational level, diagnoses, and pharmacological treatments.

Stabilized outpatients were more likely to show higher levels of PTH (B = 14.10, C.I. 0.10−28.10) (*p* < 0.05), while patients characterized by a clinical condition with psychotic symptoms presented a greater probability of low serum calcium levels (B = −0.49, C.I. −0.95−0.03) (*p* < 0.05).

Furthermore, higher BMI values proved to be negative predictors of serum levels of 25-OH-Vit D (B = −0.34, C.I −0.67–−0.01) (*p* < 0.05), as well as the number of relapses (B = −0.42, C.I. −0.79–−0.05) (*p* < 0.05). Instead, the outpatient condition (B = 12.44, C.I. 3.23−21.65) (*p* < 0.01) proved to be a positive predictor of the serum 25-OH-Vit D levels.

## 4. Discussion

The impact of acute relapses on serum levels of Vit D and PTH has not been extensively investigated among psychiatric patients. In the present study, we reported that psychiatric inpatients showed a significant difference in serum levels of both 25-OH-Vit D and PTH compared to psychopathologically and pharmacologically stabilized psychiatric outpatients. Therefore, we hypothesized that acute psychiatric relapses might represent a risk factor of low 25-OH-Vit D levels regardless of the mental diagnostic category. Indeed, correlation analysis between serum levels of 25-OH-Vit D and BPRS total and subitems scores indicated a significantly negative relationship, suggesting a possible connection between psychopathological decompensation and decreased serum levels of 25-OH-Vit D. In addition, linear regression analysis evidenced that the inpatient condition might predict low serum levels of PTH and 25-OH-Vit D.

The association between low Vit D serum levels and neuropsychiatric conditions has been the subject of recent research studies. Preclinical and clinical studies have evidenced that Vit D stimulates brain cells to produce several growth factors, such as nerve growth factor (NGF), glial cell line-derived neurotrophic factor (GDNF), and neurotrophin 3 (NT3), thus prompting the protection and growth of neuronal cells. Furthermore, the Vit D receptor, primarily expressed in the CNS, regulates behavioral and emotional responses, and modulates inflammatory responses. Vit D also decreases cytokine activity via inhibitory effects on activating and expressing inflammatory factors such as interleukins 1 and 6, tumor necrosis factor (TNF), nuclear factor kappa B (NF-kB), and a few related genes. Moreover, evidence has suggested that Vit D modulates the biosynthesis of neurotransmitters, such as serotonin through tryptophan hydroxylase 2, and neurotrophic factors, thus significantly influencing mood and causing related alterations and cognitive functions [39].

Previous studies have indicated an increased risk for Vit D deficiency in psychiatric patients, mainly those with depressive symptoms, cognitive impairment, or schizophrenia spectrum disorders [40,41,42,43,44,45]. In the present study, we included patients with several psychiatric diagnoses in the inpatient subgroup, which was different from other authors who selected patients based on a single mental disorder diagnosis [27,46,47], according to a transdiagnostic approach to mental disorders [48,49,50,51,52]. Furthermore, some cross-sectional studies reported only low serum levels of 25-OH-Vit D in hospitalized psychiatric patients [52,53,54,55,56,57], while others compared serum levels of 25-OH-Vit D of psychiatric inpatients with those of controls with no psychiatric illness [38,58,59,60,61]. In addition, other studies compared PTH and Vit D serum levels of psychiatric patients with those of healthy controls [24]. Therefore, we first compared a cohort of inpatients with ongoing acute psychiatric relapses with a psychopathologically and pharmacologically stabilized outpatient subgroup but including subjects with a prevalent mood disorder diagnosis (Table 1).

BPRS total score analysis evidenced that the inpatient subgroup had a significantly higher mean total score (*p* < 0.001) along with depression, thought disorders, activity, and hostility/suspiciousness subitem scores as well as a larger number of hospitalizations (Table 1). Moreover, correlation analysis evidenced that only 25-OH-Vit D serum levels significantly correlated negatively with total and some subitems BPRS scores. Linear regression analysis showed that the inpatient condition might represent a risk factor of low 25-OH-Vit D levels, supporting our hypothesis that the acute psychiatric relapses and not the specific psychiatric diagnosis might be a risk factor, as other studies concluded [60,61,62]. However, several studies contained no correlation analysis between low serum levels of Vit D and the scoring of psychopathological assessment scales, as evidenced by Adamson et al. (2017) in a comprehensive review on the topic [63]. Indeed, Abdullah et al. (2012), described only the association between low serum levels of Vit D and the total BPRS score of patients, presenting a cut-off score > 36 [64]. On the other hand, Menkes et al. (2012) studied the Vit D status on a sample of adult psychiatric inpatients with a prevalent diagnosis of psychotic spectrum disorders (schizophrenia, schizoaffective disorder, as well as bipolar and depressive disorders), mainly with Vit D deficiency. The authors did not report any correlation with the score of a psychopathological assessment standard scale [65].

Vit D status might be influenced by several environmental and biological factors, such as variations in sun exposure, age, sex, obesity, or chronic illnesses. In particular, low levels of 25-OH-Vit D are commonly present in obese patients. Additionally, numerous studies and the present study have evidenced the inverse association between Vit D and adiposity [66].

Indeed, most patients in our outpatient and inpatient subgroups were overweight or obese at admission. Consequently, a high BMI is to be excluded as a confounding factor in our analysis to explain the low serum concentration of 25-OH-Vit D in the inpatient cohort. Furthermore, we also confirm that a high BMI value might predict low Vit D levels, as already reported in a few studies. Other studies, instead, did not report a significant correlation between low serum Vit D levels and BMI values [29,46,53,60].

Whether Vit D deficiency is likely due to insufficient dietary intake, lifestyle habits, or environmental factors is difficult to establish in psychiatric patients. However, a significant seasonal variation of 25-OH-Vit D and sunlight exposure is now accepted as a major determinant of Vit D stores [67]. Seasonality of relapses, in particular, was significantly higher among our inpatient subgroup than in the outpatient subgroup.

In addition, the level of circulating 25-OH-Vit D in smokers is reported to be lower than that in nonsmokers [68] though some studies did not confirm such results [69]. No significant difference emerged between smokers and nonsmokers patients in our study.

Sociodemographic and biochemical factors and the genetic polymorphism of Vit D receptor, transporters, and enzymes participating in Vit D metabolism, all influencing Vit D concentration, were not examined as potential confounders of total plasma 25-OH-Vit D concentrations in our study [70].

Furthermore, we did not analyze our cohort′s comorbid medical conditions potentially associated with low Vit D serum levels (i.e., hypertension, diabetes, inflammatory bowel disease, osteoporosis, and chronic pain), which was not the aim of our study [71].

In addition, it is noteworthy to mention that magnesium also plays a crucial role in Vit D biosynthesis and metabolism and that magnesium and Vit D deficiencies may be associated with several chronic medical conditions, including psychiatric disorders. Nevertheless, in our cohort, we did not consider the biological significance of magnesium in Vit D metabolism and its therapeutic importance in minimizing complications related to Vit D deficiency [72,73,74].

Another hypothesis, suggested by in vitro studies but yet to be verified in patients, is that treatment with antipsychotic drugs might inhibit Vit D synthesis [75]. In addition, Vit D deficiency might exacerbate antipsychotic-induced metabolic disturbances [76,77,78,79]. However, establishing whether treatments with psychoactive drugs were responsible for Vit D deficiency in our inpatient and outpatient subgroups is challenging. Indeed, all patients in our cohort were on psychopharmacological treatments when plasma 25-OH-Vit D concentrations were measured.

Lastly, we need to mention a few limitations of our study, such as the relatively limited number of the enrolled participants, the heterogeneity of their clinical and psychiatric history, only one non-repeated determination of plasma levels of Vit D and PTH during hospitalization or outpatient follow-up, and BPRS administration to assess the severity of the psychopathological symptoms.

Finally, the cross-sectional observational design is to be considered since we analyzed data only at a specific time point, namely the first day of hospitalization for the inpatient subgroup and the access to day-hospital for the outpatient subgroup. Such a preliminary analysis indicated a possible association between Vit D deficiency and acute psychiatric relapses regardless of patients′ diagnostic categories. Accordingly, a longitudinal study design would appropriately explore temporal changes in the Vit D plasma levels and their impact on the psychopathological status of the included patients. In addition, a longitudinal study would enable us to collect data on variables such as exposure to sun, seasonality pattern, correlation with magnesium plasma levels, and genetic polymorphism of Vit D receptor, transporters, and enzymes participating in Vit D metabolism, which influence Vit D concentration. Our preliminary study did not examine such variables as potential confounders of the total plasma 25-OH-Vit D concentrations.

## 5. Conclusions

Our study emphasized that in psychiatric inpatients, reduced serum levels of 25-OH-Vit D and PTH might associate with the acute phase of psychiatric relapses differently than in psychiatric outpatients, who are psychopathologically and pharmacologically stabilized. Furthermore, acute psychiatric relapses might represent a risk factor for low 25-OH-Vit D levels, regardless of the mental diagnostic category. Furthermore, correlation analysis between serum levels of 25-OH-Vit D and BPRS total and subitems scores indicated a significant negative relationship, suggesting a possible connection between psychopathological decompensation and decreased serum levels of 25-OH-Vit D. In addition, linear regression analysis evidenced that the inpatient condition might be a predictor of low serum levels of PTH and 25-OH-Vit D.

Our finding of an association between acute psychiatric relapses and Vit D deficiency requires replicating in more samples and investigating to clarify the underlying mechanisms yet undefined.

Studies on the role of Vit D supplementation have produced contrasting results. Therefore, it appears uncertain whether Vit D supplements might prevent and/or treat pathological mental conditions in individuals with Vit D deficiency.

In conclusion, a comprehensive clinical practice should encompass psychosocial intervention to promote physical activity, appropriate diet, quitting smoking, and sensible sun exposure to prevent and treat Vit D deficiency in mental patients.

## Figures and Tables

**Table 1 brainsci-12-00973-t001:** Sociodemographic, clinical, and biochemical characteristics of the total number of patients with a psychiatric diagnosis, sub-grouped as inpatients and outpatients.

Variables		Total Number of Patients (n = 152)	Inpatient Subgroup (n = 74)	Outpatient Subgroup (n = 78)
**Sociodemographic**	Age at admission/first outpatient visit (mean ± sd)	47.3 ± 14.4	47.4 ± 15.1	47.1 ± 13.8
	Age at onset of the disorder (mean ± sd)	29.2 ± 12.7	27.8 ± 12.4	30.4 ± 12.9
	Gender, Male, n (%)	75 (49.3)	37 (50.0)	38 (48.7)
	Years of education (mean ± sd)	12.1 ± 4.1	10.9 ± 4.1	10.1 ± 3.7 **
	Having partner, yes, n (%)	71 (47)	22 (30)	49 (63)
	Employed, yes, n (%)	48 (32)	14 (18.9)	34 (43.6)
	Smoker, yes, n (%)	84 (55.3)	52 (70.3)	32 (41.0)
**Clinical**	Family history of psychiatric disorders, yes, n (%)	67 (48.9)	38 (52.1)	29 (45.3)
	Number of hospitalizations, (mean ± sd)	1.9 ± 2.8	2.9 ± 3.5	1.0 ± 1.5 **
	Number of relapses, (mean ± sd)	5.1 ± 7.7	1.9 ± 3.5	8.1 ± 9.3 **
	Aggressive behaviors, yes, n (%)	63 (42.9)	10 (13.5)	53 (72.6)
	Psychotic symptoms, yes, n (%)	50 (32.9)	26 (35.1)	24 (30.8)
	Suicide attempts, yes, n (%)	28 (18.5)	11 (14.9)	17 (22.1)
	Number of suicide attempts, (mean ± sd)	0.3 ± 0.8	0.3 ± 0.8	0.3 ± 0.7
	BMI (kg/m^2^), (mean ± sd)	28.0 ± 6.6	29.6 ± 7.5	26.6 ± 4.7
	Diagnosis of psychosis, yes, n (%)	25 (16.4)	25 (33.8)	0 (0) **
	Diagnosis of mood disorders, yes, n (%)	122 (80.3)	44 (59.5)	78 (100) **
	Diagnosis of OCD, yes, n (%)	5 (3.3)	5 (6.8)	0 (0) *
	BPRS, subitem and total score (mean ± sd): *Depression* *Anergy* *Thoughts disorders* *Activity* *Hostility/suspiciousness* *Total score*	13.2 ± 4.4 6.8 ± 3.5 6.7 ±3.9 4.9 ±1.8 5.5 ±2.9 36.6 ±11.4	14.9 ± 3.9 7.1 ± 4.0 8.2 ± 4.9 5.5 ± 2.1 6.6 ± 3.4 44.4 ± 11.0	11.5 ± 4.3 ** 6.5 ± 2.9 5.2 ± 1.8 ** 4.4 ± 1.3 ** 4.4 ± 1.8 ** 31.2 ±8.8 **
**Biochemical**	Calcium (mg/dL) (mean ± sd)	9.3 ± 6.4	9.2 ± 0.8	9.4 ± 1.0
	Parathormone (PTH) (pg/mL) (mean ± sd)	30.6 ± 20.1	22.1 ± 8.5	38.5 ± 24.4 **
	25-OH-Vitamin D (pg/mL) (mean ± sd)	23.1 ± 13.4	17.5 ± 8.4	28.3 ± 15.0 **

Statistically significant differences between the two subgroups: * *p* < 0.05; ** *p* < 0.001.

**Table 2 brainsci-12-00973-t002:** Pearson rho correlation coefficients of serum levels of PTH and calcium and 25-OH-vitamin D, clinical variables, and psychopathological scores.

	PTH	Calcium	25-OH-Vitamin D
**Age**	0.123	−0.062	−0.053
**Education (years)**	0.046	0.185 *	0.088
**BMI (kg/m^2^)**	−0.034	−0.126	−0.225 **
**Number of hospitalization**	−0.061	−0.016	−0.097
**Number of rehospitalization**	−0.330 **	0.029	0.063
**Suicidal attempts**	0.037	0.123	0.053
**BPRS**			
*Anxiety/depression*	−0.147	0.015	−0.172 *
*Anergy*	0.017	−0.139	−0.209 **
*Thoughts disorders*	−0.105	−0.126	−0.283 **
*Activity*	−0.063	−0.054	−0.204 *
*Hostility/suspiciousness*	−0.128	−0.018	−0.187 *
*Total score*	−0.158	−0.108	−0.322 **

* *p* < 0.05; ** *p* < 0.01.

**Table 3 brainsci-12-00973-t003:** Predictors of serum PTH, calcium, and 25-OH-vitamin D levels according to adjusted multinomial logistic regression model.

	Dependent Variable					
Independent Variable	PTH		Calcium		25-OH-Vitamin D	
	B (95% C.I.)		B (95% C.I.)		B (95% C.I.)	
**BMI (kg/m^2^)**	0.22	(−0.29–0.72)	−0.00	(−0.03–0.02)	−0.34	(−0.67–−0.01) *
**Total number of hospitalization**	−0.34	(−1.71–1.03)	0.27	(−0.04–0.09)	0.72	(−0.18–1.62)
**Number of relapses**	0.51	(−0.06–1.08)	−0.00	(−0.03–0.03)	−0.42	(−0.79–−0.45) *
**Number of suicidal attempts**	−5.09	(−12.09–1.91)	0.22	(−0.12–0.56)	3.04	(−1.56–7.64)
**History of suicidal attempts**	9.51	(−4.15–23.18)	−1.60	(−0.82–0.50)	−2.33	(−11.32–6.655)
**Outpatients, yes**	14.10	(0.10–28.10)*	0.19	(−0.49–0.86)	12.44	(3.23–21.65) **
**Having partner, yes**	−1.68	(−4.50–1.15)	−0.16	(0.15–0.12)	−1.06	(−2.92–0.80)
**Employment, yes**	2.82	(−2.28–7.91)	−0.12	(−0.36–0.13)	−1.64	(−4.99–1.71)
**BPRS total**	−0.06	(−0.42–0.30)	−0.00	(−0.02–0.01)	−0.12	(−0.34–0.12)
**Age at onset**	0.02	(−0.31–0.34)	0.00	(−0.01–0.02)	0.07	(−0.14–0.29)
**Aggressive behavior**	3.66	(−5.03–12.35)	−0.15	(−0.27–0.57)	−2.68	(−8.39–3.04)
**Psychotic symptoms**	4.88	(−4.71–14.47)	−0.49	(−0.95–0.03) *	2.31	(−3.92–8.62)
**Seasonality**	2.31	(−8.09–12.71)	0.08	(0.42–0.58)	1.22	(−5.61–8.06)

Model was adjusted for gender, age at onset and admission, marital status, educational level, diagnosis, and pharmacological treatments. * *p* < 0.05; ** *p* < 0.01.

## Data Availability

Data presented in this study are available in the tables; clinical data of the included patients are available at the Department of Psychiatry, Department of Mental Health and Public Medicine, University of Campania “Luigi Vanvitelli”, Naples.

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
