# Peer review of "Vitamin D Deficiency and Risk Factors Related to Acute Psychiatric Relapses in Patients with Severe Mental Disorders: A Preliminary Study"

_brainsci, 2022, doi:10.3390/brainsci12080973_

Round 1

Reviewer 1 Report

The manuscript entitled ‘Vitamin D deficiency and Risk Factors Related to Acute Psychiatric Relapses in Mental Patients’ presents interesting issue, however major corrections are needed.

 Introduction:

      Lines 97-99  – ‘Subjects were classified as normal weight (BMI 18.50–24.99 kg/m2 ), overweight (BMI 25.00–29.99 kg/m2 ), and obese (BMI ≥ 30.00 98 kg/m2)’ – what about underweight BMI <18.5 ?

      Lines 100-101 – ‘Psychopathological symptom severity was assessed by the Brief Psychiatric Rating 100 Scale (BPRS, 18-item version),’ – More information is needed about the validity and reliability of each measure. Additionally, any limitations in reliability and validity need to be addressed in the discussion.

      There are some problems with BMI value – in table 1 BMI (kg/m2) for total is 39.0 ± 25.7, but for sub-groups are 29.6 ± 7.5 and 26.6 ± 4.7!!! Please check it. This is a crucial issue due to the fat, that the prevalence of vitamin D deficiency is highest in individuals with higher BMI.

      Lines 201-202 –‘ In the present study, we report that psychiatric inpatients showed a significant reduction in serum levels of both 25-OH-Vit D and PTH compared to psychopathologically and pharmacologically stabilized psychiatric outpatients.’ There was no ‘reduction in serum levels’ due to the fact, that there were no observations in time. There was significant 'differences in serum levels' … please be more specific.

   In discussion session information on potential mechanisms/relationship of vitamin D and mental health is missing.

  Conclusion “Psychiatric patients might be more inclined to develop low levels of Vit D due to diminished outdoor activity, reduced nutrient intake, and psychopharmacological drug treatments.” – this aspects were not measured. If so, please provide data from three-day dietary record/ FFQ or 24h recall.

      The results should be related only to the obtained data.

Reviewer 2 Report

This is a good study except for one thing:  you did not include measure of magnesium status which has been shown to impact vitamin D levels of serum as well as all the conditions you cite in your Introduction that are associated with vit D status, including those associated with the central nervous system.   It is really important that studies on vit D deficiency and psychiatric disorders include an assessment of magnesium status.    This is a sad default of your study, which is good in and of itself.  But, I suggest that you could add to your discussion this important omission and show at least some of the published work on this subject, which goes back years.   Here are a few for you to consider studying and hopefully including in your next study.  There are more.  

1. Reddy P, Edwards LR. Magnesium Supplementation in Vitamin D Deficiency. Am J Ther 2017 doi: 10.1097/MJT.0000000000000538 [published Online First: 2017/05/05].  https://www.ncbi.nlm.nih.gov/pubmed/28471760  

2. Uwitonze AM, Razzaque MS. Role of Magnesium in Vitamin D Activation and Function. J Am Osteopath Assoc 2018;118(3):181-89. doi: 10.7556/jaoa.2018.037 [published Online First: 2018/02/27].  https://www.ncbi.nlm.nih.gov/pubmed/29480918  

3. Peeri NC, Egan KM, Chai W, et al. Association of magnesium intake and vitamin D status with cognitive function in older adults: an analysis of US National Health and Nutrition Examination Survey (NHANES) 2011 to 2014. Eur J Nutr 2020 doi: 10.1007/s00394-020-02267-4 [published Online First: 2020/05/11].  https://www.ncbi.nlm.nih.gov/pubmed/32388734  

Round 2

Reviewer 1 Report

 I appreciate the great efforts that the authors have made in response to my questions and concerns. However, there are some technical issues with the manuscript (e.g. table 1) that must be solved in proofs.

Author Response

As the reviewer reported, the technical problems of Table 1 will be resolved when proofs of the manuscript will be available.

We thank the reviewer very much for appreciating our efforts.